# Analysis of Attendance and Speleotourism Potential of Accessible Caves in Karst Landscape of Slovakia

**Vladimír Čech** [1] , **Peter Chrastina** [2], **Bohuslava Gregorová** [3,*] , **Pavel Hronček** [3], **Radoslav Klamár** [1] **and Vladislava Košová** [1]

1   Department of Geography and Applied Geoinformatics, Faculty of Humanities and Natural Sciences, University of Prešov, Ulica 17. Novembra 1, 081 16 Prešov, Slovakia; vladimir.cech@unipo.sk (V.Č.); radoslav.klamar@unipo.sk (R.K.); vladislava.kosova@smail.unipo.sk (V.K.)
2   Department of Historical Sciences and Central European Studies, Faculty of Arts, University of Ss. Cyril and Methodius Trnava, Námestie J. Herdu 2, 917 01 Trnava, Slovakia; peter.chrastina@ucm.sk
3   Department of Geography and Geology, Faculty of Natural Sciences, Matej Bel University, Tajovského 40, 974 01 Banská Bystrica, Slovakia; pavel.hroncek@umb.sk
*   Correspondence: bohuslava.gregorova@umb.sk

**Abstract:** Caves represent natural phenomena that have been used by man since ancient times, first as a refuge and dwelling, and later as objects of research and tourism. In the karst landscape of Slovak Republic in Central Europe, more than 7000 caves are registered in a relatively small area, of which 18 are open to the public. This paper deals with the analysis of the speleotourism potential of 12 of these caves, administered by the Slovak Caves Administration. Based on the obtained data, we first evaluate the number of visitors in 2010–2019. Using a public opinion survey among visitors, we then evaluate the individual indicators of quality and each cave's resulting potential. We use a modified standardization methodology and standardization of individual evaluation criteria weights for individual evaluation indicators. The resulting values of the potential of caves for speleotourism point to the great importance of these sites for domestic and foreign tourism and the protection of nature and landscape, as 5 of these caves have been part of the UNESCO World Natural and Cultural Heritage List since 1995.

**Keywords:** cave; karst landscape; cave attendance; speleotourism; indicators of quality

## 1. Introduction

Many structures of a natural character have considerable tourism potential. These include mountains, canyons, islands, beaches, rivers, waterfalls, flora, and fauna. Among these vast possibilities, caves stand out primarily due to their unique properties, both scientific and aesthetic, which result in their high degree of attraction [1]. A cave represents all genetic types of underground spaces that can be explored by man, entirely or partially surrounded by rock, spanning from abri through a fissure, river caves to multilevel combined cave systems [2]. The National Council of the Slovak Republic Act no. 543/2000 Coll. on nature and landscape protection defines a cave as a hollow underground space in the Earth's crust accessible to humans and created by natural processes, the length or depth of which exceeds 2 m and the dimensions of the surface opening are less than its length or depth. The majority of caves in Slovakia are associated with the specific environment of the karst [3].

The economic benefits of speleotourism are considerable. Profits obtained directly and indirectly from caves can become very important for tourism at the local level [4,5]. However, insufficient regulation of attendance, maintenance, or infrastructure management, in general, can result in a severe threat to the underground environment and the development of speleotourism itself [6,7]. Gurnee R. and Gurnee J. [8] presented a combination of four crucial factors for the successful development and operation of a cave used for tourism.

These factors are scientific research, art, technology, and management. Cigna and Burri [9] recommend scientific studies at the beginning of the first cave research phase. Design is needed in determining the direction of sidewalks and in selecting cave scenes. Lighting is a combination of art and another factor, technology. Technology is vital for regulating water and other natural forces in the cave and for designing suitable routes. Management is a process that continues from the moment the first established plans are developed and operational. Lobo et al. [1] point out that the management of tourist activities in protected cave areas requires quality and scientifically based knowledge of the environment, emphasizing the limits of the resilience of geo- and ecosystems directly and indirectly affected by such activities.

When appropriately used, caves are essential for tourism development since they can increase the economic, social, and environmental benefits for the host community relatively quickly. However, the development of tourism has both positive and negative impacts on the local community. These are the economic, socio-cultural, and environmental consequences that result from interactions between the local community and visitors during tourist experiences or meetings. Tourism as a business activity can bring economic, social, and environmental changes to the destination country of tourism [10].

Analysis of the interactions in the human–karst relationship with respect to speleotourism in relation to its sustainability and the proper management of show caves is an area that has been addressed by several authors in recent years [11–23].

The presented paper's primary goal is to analyze the number of visitors and the potential of caves for speleotourism using the obtained data and the results of a public opinion survey among visitors of 12 accessible caves in Slovakia, which belong to the Slovak Caves Administration. We analyzed attendance of these caves in the summer months of 2010–2019 and conducted a public opinion survey. We use a modified standardization methodology and standardization of individual evaluation criteria weights for individual evaluation indicators. We selected 15 criteria for evaluating the natural potential of caves. We assigned a certain weight to each criterion according to its importance for the potential of caves. The weights were based on the evaluation of students in a public opinion survey and on the basis of literature adapted to the conditions of Slovakia. The resulting values of caves' potential for speleotourism point to their differentiated significance in tourism development.

Using the knowledge from previous research in this area, evaluation of the development, and changes in the number of visitors to individual caves, we were looking for answers to the following research hypotheses:

1.  We assume the highest cave attendance in 2019 is due to the continuous increase in the number of tourists in Slovakia since 2014, culminating in 2019.
2.  We expect the highest number of visitors in the most famous caves of the Western Carpathians located in national parks, namely Demänovská Cave of Liberty in the Low Tatras National Park and Belianska Cave in the Tatra National Park.
3.  We assume that the approach and interpretation of the cave guide are essential factors increasing the evaluation of the attractiveness of caves in terms of the potential for speleotourism.
4.  We assume a correlation between the highest number of visitors and the highest value of the natural potential of caves for the development of speleotourism and, similarly, a correlation between the lowest number of visitors and the lowest natural potential of caves for the development of speleotourism.
5.  We assume that the Demänovská Cave of Liberty has the highest potential for speleotourism in terms of natural attractions and attendance.

## 2. Materials and Methods

### 2.1. Theoretical Background

According to [24], the cave is one of the first documented geological phenomena that became the subject of tourism. Some authors believe that visiting caves is essentially the

oldest form of geotourism [25]. Geotourism is a relatively recent phenomenon based on an old idea and belongs to the specific interest tourism category. Simply, geotourism is the promotion and protection of geological heritage through tourism by education and explanation [26]. Similarly, Slomka and Kicińska-Świderska [27] state that geotourism is the knowledge of geological objects and processes. It is a sustainable form of tourism with a primary focus on using the Earth's geological features and creating the products of geotourism [28]. The following authors of [9,28,29] dealt with the definition, specifics, and characteristics of geotourism, among others. Newsome, Dowling [30] state that, unlike ecotourism which, by definition, can only occur in areas of natural character, geotourism can take place in both natural and human-made areas (e.g., in mining areas within the so-called mining tourism [29,31–33]). One of the particular forms of geotourism, which focuses on the knowledge of specific geological objects and endokarst character processes, is cave tourism —speleotourism (speleotourism as a form of geotourism, e.g., in [34–40]). Speleological objects represent important symbols of geological and geomorphological processes globally and are highly attractive to many tourists. This form of tourism has an exceptional and unique position in the world of tourism [41]. Similarly, Knežević and Grbac-Žiković [42] state that caves are referred to as speleological objects and the tourism based on them is called speleotourism or cave tourism. Pásková and Zelenka [43] understand speleotourism as a form of tourism associated with exploring and discovering cave spaces and abysses. It usually includes climbing activities (abseiling, climbing ascents), cave diving, and boat trips on underground rivers with guides. According to [44], speleotourism is an organized speleological activity focused on cultural and cognitive activities in inaccessible endokarst objects (caves, abysses, and others), available to the interested in essential equipment (helmet, own lighting) and under the guidance of an experienced professional guide (speleologist—conservationist).

These definitions are examples of a narrower understanding of the term speleotourism. Speleotourism, in the broadest sense of the word, is understood as an individual or group tourism organized in the commercial interest in caves accessible to the public or other attractive karst objects equipped with security features. These objects are illuminated and have marked paths, stairs, and locked entrances, and some have electric lifts. They are technically secured, and a trained guide carries out the tour. As a form of tourism, cave tourism receives increased attention especially in the countries where the limestone (karst) relief is widespread (National Association of Caves [45]). This natural landscape is fascinating and valuable for tourists, and caves are crucial elements of this karst system, intervening in transforming the physical environment on and below the surface [46]. Other authors [47,48] also point to the use of karst areas and especially caves as their unique elements with considerable tourist attractiveness for the development of tourism and recreation. In this context, Rindam [49] also states that cave tourism is becoming more important for tourism development.

### 2.2. Study Area

According to [50,51], the karst areas of Slovakia cover an area of approximately 2700 km$^2$ with over 7000 caves registered in this small area. A total of 18 of these are open to the public. Thirteen of them (Driny Cave, Harmanecká Cave, 150 Brestovská Cave, Demänovská Cave of Liberty, Demänovská Ice Cave, Bystrianska Cave, Važecká Cave, Belianska Cave, Dobšinská Ice Cave, Ochtinská Aragonite Cave, Gombasecká Cave, Domica Cave, Jasovská Cave) are managed by Slovak Caves Administration seated in Liptovský Mikuláš, which is a contributory organization of the State Nature Conservancy of the Slovak Republic. Another entity manages five caves (Malá Stanišovská Cave, Cave of Dead Bats, Krásnohorská Cave, Zlá diera Cave, and Bojnice Castle Cave). For the paper, we have designated caves in the Slovak Caves Administration except for the Brestovská Cave, accessible to the public only for a short time (since 2016) (Figure 1, Table 1).

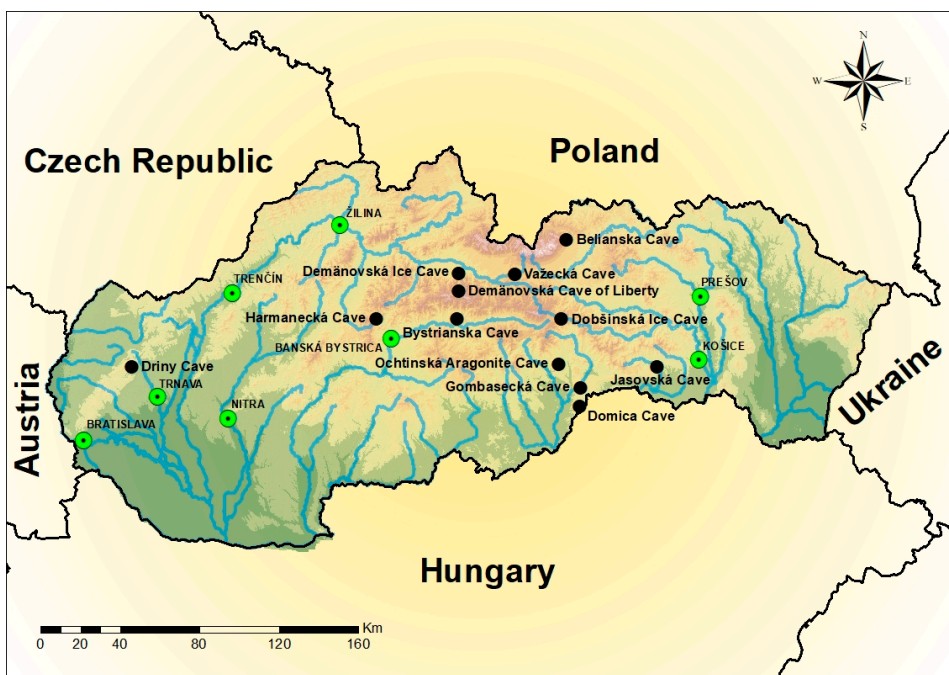

**Figure 1.** Location of studied caves within the Slovak Republic.

Belianska Cave is the only classically accessible cave in the High Tatras National Park. The interior of the Belianska Cave is made unique by pagoda-shaped stalagmites (Palm Hall), sinter waterfalls (SNP Dome, Music Hall), and other forms of sinter filling (Klenotnica, Gallery). The cave entrance areas were already known to gold diggers in the first half of the 18th century, but it was not until J. Husz and J. Britz that penetrated the underground spaces in 1881, and a year later, the cave was opened to the general public. Belianska Cave has been electrically lit since 1896 [52,53].

As part of the Horehronské podolie geomorphological unit, unique underground spaces of the Bystrianska Cave were formed on the southern edge of the village Bystrá. The sinter forms in the Bystrianska Cave are mainly sinter curtains (Kaplnka) hanging, in some cases, from the edges of sinter crusts formed on later melted fluvial sediments (Baldachýn). Rare aragonite can be found in the area delimited by the Peklo abyss. The first daredevils who successfully entered the underground labyrinth were J. Kovalčík and E. Laubert in 1923 [50,54,55].

In the Demänovská Dolina National Nature Reserve, located at the northern foot of the Low Tatras, two caves are accessible in a classical way, the first of which is the Demänovská Cave of Liberty. We consider this cave to be the most morphologically diverse part of the more than 43 km long Demänovský Cave System, representing the longest cave system in Slovakia. The visitor is attracted by the sight of massive sinter waterfalls and stalagmites, spherulitic stalactites, and many other forms. The cave was discovered in 1921, when A. Král, with the help of A. Mišura and other explorers, discovered it with the dry lowest dive of Demänovka and named it the Chrám slobody (Temple of Liberty) [50,56–59].

The second cave accessible to the public in the Demänovská Dolina National Nature Reserve is the Demänovská Ice Cave, located 10 km south of Liptovský Mikuláš. Visitors will encounter the ice fill, especially in the Kmeť Dome, located in the cave's cooler lower parts. This specific type of cave decoration consists mainly of floor ice, stalactites, stalagmites, and ice pillars. However, in recent years, the ice decoration has been significantly reduced, and during some years, only floor ice remained. According to undocumented sources, the cave spaces have been known since ancient times, but the first written mention by J.P. Hain dates back to 1672, who studied the bones of cave bears [50,56–59].

**Table 1.** Basic parameters of the studied caves.

| | Entrance Elevation (m a.s.l.) | Ascent to the Cave from the Parking Lot (min) | Overall Cave Length (m) | Length of the Tour (m) | Elevation Difference of the Tour (m) | Number of Stairs on the Tour | Time Needed to Complete the Tour (min) | Ticket Price for Adult Person in € (2020) |
|---|---|---|---|---|---|---|---|---|
| Belianska Cave | 890 | 25 | 3829 | 1370 | 125 | 874 | 70 | 9 |
| Bystrianska Cave | 565 | 1 | 3531 | 580 | 0 | 26 | 45 | 6 |
| Demänovská Cave of Liberty | 870 | 15 | 11,117 | 1150 (traditional tour) 2150 (long tour) | 86 (both tours) | 913 (t.t.) 1 118 (l.t.) | 60 (traditional tour) 100 (long tour) | 9 (traditional tour) 16 (long tour) |
| Demänovská Ice Cave | 840 | 20 | 2445 | 650 | 48 | 670 | 45 | 8 |
| Dobšinská Ice Cave | 969.5 | 25 | 1491 | 515 | 43 | 500 | 30 | 8 |
| Domica Cave | 339 | 1 | 5368 | 780 (short tour) 930 (tour with a boat ride) | 7 | ? | 45 (short tour) 60 (tour with a boat ride) | 6 (short tour) 8 (tour with a boat ride) |
| Driny Cave | 399 | 20 | 680 | 450 | 10 | 151 | 35 | 6 |
| Gombasecká Cave | 250 | 2 | 1525 | 530 | 8 | 87 | 30 | 6 |
| Harmanecká Cave | 821 | 40 | 3123 | 1020 | 64 | 1042 | 60 | 7 |
| Jasovská Cave | 257 | 1 | 2811 | 720 | 30 | 339 | 45 | 6 |
| Ochtinská Aragonite Cave | 642 | 10 | 585 | 300 | 19 | 104 | 30 | 7 |
| Važecká Cave | 784 | 1 | 530 | 235 | 5 | 88 | 25 | 5 |

The Dobšinská Ice Cave National Nature Monument (in National Park Slovak Paradise) is one of the most important ice caves globally regarding its location outside the Alpine region, which is inscribed on the UNESCO World Cultural and Natural Heritage List since 2000. The cave is a part of the 23 km long Stratenská Cave Complex. The cave's central part is a massive cavity extending from the surface opening to a depth of 70 m, which is mostly filled with ice dividing the cave into separate spaces (Prízemie, Malá sieň, Veľká sieň). The cave opening has been known as an "ice hole" since ancient times, but E. Ruffíny, accompanied by other discoverers, was the first to descend into the cave in 1870. The cave was made accessible to the public one year later. The operators managed to provide electric lighting in 1887, and the Dobšinská Ice Cave became one of the first electrified caves in the world [60–62].

The most famous and longest cave of the Slovak Karst National Park, inscribed on the prestigious UNESCO World Cultural and Natural Heritage List, is the Domica Cave, which attracts visitors with its exceptional archeological finds and numerous sinter drums and shields. Domica Cave (Slovakia) and Baradla Cave (Hungary) form a complex genetic unit. The uniqueness of Domica is the existence of an underground canyon with meanders, modeled by the underground river Styx. The cave spaces were shortly used by the oldest Neolithic inhabitants with eastern Linear Pottery Culture, but Neolithic people of the Bukovohorská culture prevailingly inhabited it for an extended period. In 1926, J. Majko climbed into the massive cave spaces of Domica, where several diverse archaeological finds, including arrows, combs, rings, axes, knives, or scrapers, were found [63–65].

The only accessible cave in the western part of Slovakia is the Driny Cave, located in the Smolenice karst, whose territorial protection is provided by the Malé Karpaty Protected Landscape Area. Narrow crack corridors form most of the underground spaces (Beňovského chodba, Chodba nádejí a spolupracovníkov) accompanied by smaller hall spaces (Sieň Slovenskej speleologickej spoločnosti) created at the connection of the tectonic faults. J. Banič and I. Vajsábel did not get into the deeper parts of the sinkhole chimney leading to the cave until 1929 [66].

The Slovak Karst National Park territory is dotted with many cave spaces, among which the Gombasecká Cave, included in the UNESCO World Heritage List due to the characteristic occurrence of thin tubular stalactites, called sinter quills, plays an important role. The cave, together with the Silická ľadnica Cave, forms the Silicko-Gombasecký subsurface hydrological complex. Gombasecká Cave is famous for its unique thin sinter straws, growing to a length of 3 m. The first discoverers of the Gombasecká Cave were the voluntary cavers from Rožňava, who managed to enter the underground spaces in 1951 through Čierna vyvieračka [67,68].

The Harmanecká Cave, with massive domes filled with rich sinter decoration and an inseparable part of a large colony of wintering bats, is situated in the southern part of the Great Fatra National Park, in the Harmanecká Valley. The cave is known for the abundant occurrence of white soft sinter—moonmilk, from which sinter lakes, wall flowstone waterfalls and draperies, and pagoda-shaped stalagmites, protruding to a height of up to 12 m, were formed over a more extended geological period. The cave discoverer's status was granted to an 18-year-old man Michal Bacúrik, who in 1932, took part in digging the so-called Discoverer's Passage leading to the snow-white hall, called the Discoverer's Dome [69].

The third classically accessible cave in the Slovak Karst National Park is the Jasovská Cave, which has gained a UNESCO protected monument's status thanks to various calcite sinter filling types, a unique rock formation, and numerous archaeological finds. According to the documented legends, it is said that Premonstratensian Jasov monks discovered the cave during the 12th century. With the strong support of Alojz Richter, the order of the Premonstratensian canons in Jasov, the cave was opened to the public for the first time in 1846 [70–72].

In the eastern part of the Revúcka Highlands, also known as the Ochtinský crypto-karst, are unique underground spaces of the Ochtinská Aragonite Cave, listed on the

UNESCO World Cultural and Natural Heritage List. The cave can be described by a unique three-generation aragonite decoration, while the oldest karst forms are turbid kidney formations and their corroded remains. The prominent representatives are spiral helictites, or tufted or bushy formations, attracting many visitors' attention. Its history began to be written in 1954 when it was accidentally discovered by employees of the Eastern Slovakia Ore Survey in Jelšava [73–77].

At the junction of the Kozie chrbty ridge with the Liptovská kotlina Basin, the Važecká Cave stretches in the Važecký Karst, on the western edge of the village of Važec. Typical forms of cave decoration include stalagmites, stalactites, and sinter lakes. The occurrence of fine sediments, washed up by the former floodwaters of Biely Váh, was confirmed in many parts of the cave. The residents of Važec have known the entrance hall of Važecká Cave for a long time. The first discoverers of the more distant parts of the cave were O.A. Húska and A. Somra in 1922 [50,78,79].

### 2.3. Methods

We analyzed the attendance in the studied caves based on the Administration of Slovak Caves' data. We took into account a time interval of ten years (2010–2019), which we evaluated as sufficient to capture the trends and development dynamics in this area. We focused on three months in the primary tourist season—June, July, and August, as only during these summer months all 12 caves are opened to the public. The analysis of attendance is also essential compared to the final evaluation of caves' potential for speleotourism. We were especially interested in whether the highest number of visitors would be shown in the caves with the highest potential.

Given the study's primary goal—to determine the attractiveness of caves for speleotourism—we decided to conduct a public opinion survey regarding the general methodology of questionnaire creation [80,81]. We have selected 15 essential criteria for evaluating the potential. The respondents' task was to determine the order of significance of these criteria (classification indices) in the first step. Each criterion was scored according to significance from 1 (least significant) to 5 (most significant). The criteria were divided into five significance classes (Figure 2, Table 2). Importance of the classes were rated according to studies [82–85] modified to the conditions of the Slovak Republic.

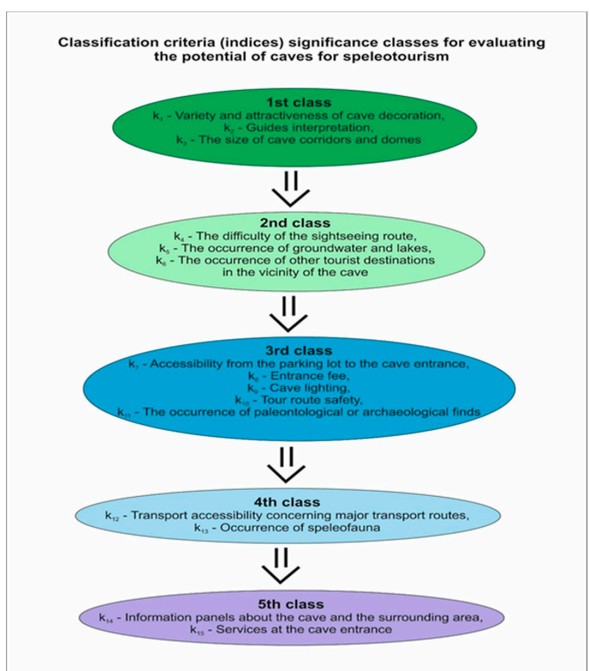

**Figure 2.** Classification criteria (indices) significance classes for evaluating the potential of caves for speleotourism (the 1st class—most significant criteria, the 5th class—least significant criteria).

**Table 2.** Order of individual criteria significance and calculation of their weight value.

| Classification Index (Criterion) | $k_{r_i}$ | $k_{r_i}^s$ | $W_j$ | $W_j^n = W_j^n/y$ |
|---|---|---|---|---|
| $k_{1_i}$ Variety and attractiveness of cave decoration | 1 | 2 | 14.5 | 0.104 |
| $k_{2_i}$ Guide interpretation | 1 | 2 | 14.5 | 0.104 |
| $k_{3_i}$ The size of cave corridors and domes | 1 | 2 | 14.5 | 0.104 |
| $k_{4_i}$ The difficulty of the sightseeing route | 2 | 5 | 12.5 | 0.089 |
| $k_{5_i}$ The occurrence of groundwater and lakes | 2 | 5 | 12.5 | 0.089 |
| $k_{6_i}$ The occurrence of other tourist destinations in the vicinity of the cave | 2 | 5 | 12.5 | 0.089 |
| $k_{7_i}$ Accessibility from the parking lot to the cave entrance | 3 | 9 | 9 | 0.064 |
| $k_{8_i}$ Entrance fee | 3 | 9 | 9 | 0.064 |
| $k_{9_i}$ Cave lighting | 3 | 9 | 9 | 0.064 |
| $k_{10_i}$ Tour route safety | 3 | 9 | 9 | 0.064 |
| $k_{11_i}$ The occurrence of paleontological or archaeological finds | 3 | 9 | 9 | 0.064 |
| $k_{12_i}$ Transport accessibility concerning major transport routes | 4 | 12.5 | 5 | 0.036 |
| $k_{13_i}$ Occurrence of speleofauna | 4 | 12.5 | 5 | 0.036 |
| $k_{14_i}$ Information panels about the cave and the surrounding area | 5 | 14.5 | 2 | 0.014 |
| $k_{15_i}$ Services at the cave entrance | 5 | 14.5 | 2 | 0.014 |

Explanations: $k_{r_i}$—simple order of significance, $k_{r_i}^s$—standardized index order, $W_j$—the weight of indices, $W_j^n$—the standardized weight of indices, y = (W1 + W2 + W3 + ... + W15).

Subsequently, the index's standardized order and weight were determined, and the index's standardized weight was calculated according to [86,87]. The methodology was also used to determine the standardized order of the index and its weight and calculate the value of the potential for individual classification criteria (indices) (as an example, Table 3). These criteria were graded with different potential values. Subsequently, the respondents filled in individual tables of criteria for each cave separately. Within each classification index, the final value for the classification index was obtained, and the final value of the cave potential (K) for speleotourism was also calculated based on the formula:

$$K = \sum_{i=1}^{3} K_{1_i} \cdot W_1^n + \sum_{i=1}^{3} K_{2_i} \cdot W_2^n + \sum_{i=1}^{3} K_{3_i} \cdot W_3^n + \sum_{i=1}^{3} K_{4_i} \cdot W_4^n + \sum_{i=1}^{3} K_{5_i} \cdot W_5^n + \sum_{i=1}^{3} K_{6_i} \cdot W_6^n$$
$$+ \sum_{i=1}^{3} K_{7_i} \cdot W_7^n + \sum_{i=1}^{2} K_{8_i} \cdot W_8^n + \sum_{i=1}^{3} K_{9_i} \cdot W_9^n + \sum_{i=1}^{3} K_{10_i} \cdot W_{10}^n + \sum_{i=1}^{3} K_{11_i} \cdot W_{11}^n + \sum_{i=1}^{3} K_{12_i} \cdot W_{12}^n$$
$$+ \sum_{i=1}^{3} K_{13_i} \cdot W_{13}^n + \sum_{i=1}^{3} K_{14_i} \cdot W_{14}^n + \sum_{i=1}^{3} K_{15_i} \cdot W_{15}^n$$

**Table 3.** Example of Classification index: Variety and attractiveness of cave decoration.

| Classification Index $k_{1_i}$ | $k_{1_i}$ | $k_{1_i}^s$ | $K_{1_i}$ | $K_{1_i} \cdot W_1^n$ |
|---|---|---|---|---|
| Rich occurrence of dripstone (or ice) decoration along the entire tour route | 1 | 1 | 3 | 0.312 |
| Sporadic occurrence of dripstone (or ice) decoration in some corridors and halls | 2 | 2 | 2 | 0.208 |
| Low occurrence of decoration, a predominance of spaces without dripstone (or ice) decoration | 3 | 3 | 1 | 0.104 |

Explanations: $k_{1_i}$—simple order of significance, $k_{1_i}^s$—standardized index order, $K_{1_i}$—the weight of index.

The respondents were 200 visitors to the caves. The majority were students of the University of Prešov, Faculty of Humanities and Natural Sciences (112). Students of the Department of Geography and Applied Geoinformatics (44) completed routes in all localities en masse as part of field practice. Other students of the University of Prešov in the fields of history, ecology, German language, Slovak language, Ukrainian language,

and biology visited caves individually, within their free time. All students were informed in detail that they would participate in the public opinion survey only after completing the visit to the caves. Their participation in the public opinion survey, as with other respondents, was voluntary. The group of other respondents, numbering 88, consisted of individual visitors to the examined caves, whom the authors of the study addressed after completing a tour of the cave. These were mainly families with children, and only adults took part in the survey.

Another classification indexes ($k_{2_i}$–$k_{15_i}$):

Interpretation of the guide—Sufficient, funny, with the opportunity to ask questions (0.312); Average, basic information without the possibility of asking additional questions (0.208); Insufficient, little information (0.104).

The size of cave corridors and domes—The predominance of massive and large spaces (0.312); Local occurrence of massive and large spaces (0.208); Low incidence of massive and large spaces, the dominance of narrow crevice passages (0.104).

The difficulty of the sightseeing route—Easy, without significant elevations (0.267); Moderately demanding, the sporadic occurrence of stairs (0.178); Challenging, higher elevation, frequent occurrence of stairs (0.089).

The occurrence of groundwater and lakes—The occurrence of water flow and lakes on the tour route (0.267); The occasional occurrence of lakes on the tour route (0.178); Without watercourses or lakes on the tour route (0.089).

The occurrence of other tourist destinations in the vicinity of the cave—The occurrence of other tourist destinations within a distance of 10 km from the cave (0.267); The occurrence of other tourist destinations within a distance of 20 km from the cave (0.178); The occurrence of other tourist destinations within a distance of 50 km from the cave (0.089).

Accessibility from the parking lot to the cave entrance—Easy (0.192); Average (0.128); Difficult (0.064).

Entrance fee—Proportionate to the acquired experience (0.128); Disproportionate to the acquired experience (0.064).

Cave lighting—Sufficient—suitably complements the tour of the cave (0.192); On average—a different intensity would be appropriate in some areas (0.128); Insufficient—poor lighting (0.064).

Tour route safety—Sufficient—sufficient occurrence of security elements—stairs, railings, and others (0.192); On average—in some areas, it would be appropriate to place more security features (0.128); Insufficient—weak occurrence of security elements (0.064).

The occurrence of paleontological or archaeological finds—A rich occurrence of paleontological or archaeological finds (0.192); A rare occurrence of paleontological or archaeological finds (0.128); No paleontological or archaeological finds (0.064).

Transport accessibility concerning major transport routes—Good (0.108); Average (0.072); Bad (0.036).

Occurrence of speleofauna—A rich occurrence of speleofauna on the tour route (0.108); Sporadic occurrence of speleofauna on the tour route (0.072); No speleofauna on the tour route (0.036).

Information panels about the cave and the surrounding area—Sufficient occurrence (0.042); Average occurrence (0.028); Insufficient occurrence (0.014).

Services at the cave entrance (souvenirs, toilets, and others)—Sufficient level of service (0.042); An average level of service (0.028); Insufficient level of services (0.014).

## 3. Results

In terms of attendance analysis, we focused on the summer months of 2010–2019, when all 12 caves are open to the public (Table 4).

**Table 4.** Total attendance of studied caves during the summer months for the period 2010–2019.

| Cave | 2010 | 2011 | 2012 | 2013 | 2014 | 2015 | 2016 | 2017 | 2018 | 2019 |
|---|---|---|---|---|---|---|---|---|---|---|
| Belianska Cave | 55,496 | 63,120 | 60,402 | 64,801 | 69,056 | 64,662 | 77,086 | 76,902 | 84,171 | 87,544 |
| Bystrianska Cave | 12,298 | 12,597 | 12,994 | 12,180 | 13,942 | 14,632 | 15,281 | 16,842 | 17,982 | 17,821 |
| Demänovská Cave of Liberty | 58,195 | 63,118 | 59,845 | 61,403 | 61,692 | 62,337 | 76,090 | 69,174 | 67,074 | 68,902 |
| Demänovská Ice Cave | 56,367 | 37,267 | 60,085 | 62,889 | 62,161 | 62,468 | 66,250 | 66,710 | 64,241 | 62,072 |
| Dobšinská Ice Cave | 45,120 | 48,606 | 49,438 | 48,890 | 52,581 | 57,868 | 67,024 | 72,174 | 68,384 | 68,681 |
| Domica Cave | 17,554 | 18,537 | 11,614 | 16,371 | 13,298 | 13,646 | 27,835 | 19,583 | 19,111 | 16,083 |
| Driny Cave | 21,323 | 21,461 | 24,880 | 24,795 | 23,946 | 24,744 | 27,518 | 28,277 | 26,730 | 26,256 |
| Gombasecká Cave | 5863 | 6056 | 6125 | 5057 | 6153 | 6609 | 8283 | 9505 | 8645 | 10,629 |
| Harmanecká Cave | 11,076 | 14,342 | 14,976 | 13,988 | 13,880 | 15,846 | 16,345 | 15,857 | 15,148 | 16,166 |
| Jasovská Cave | 7712 | 9947 | 10,491 | 11,022 | 11,684 | 12,496 | 11,683 | 14,532 | 12,719 | 14,773 |
| Ochtinská Aragonite Cave | 16,407 | 17,127 | 16,165 | 14,319 | 17,112 | 18,925 | 20,575 | 24,193 | 22,110 | 25,253 |
| Važecká Cave | 10,270 | 10,678 | 9859 | 9893 | 11,080 | 11,183 | 12,265 | 12,502 | 13,172 | 12,019 |
| SUM | 317,681 | 322,856 | 336,874 | 345,608 | 356,585 | 365,416 | 426,235 | 426,251 | 419,487 | 426,199 |

Data source: Statistical data of the Slovak Caves Administration, Liptovský Mikuláš.

In terms of the total number of visitors, we divided the examined caves into three categories:

The first category (most visited caves)—caves with a total attendance of over 500,000 visitors during the three summer months for the period 2010–2019. The above statistical data show that the most visited cave in Slovakia is the Belianska Cave. The cave's total attendance for the three summer months reached 703,240 visitors (2010–2019). The second most visited cave is the Demänovská Cave of Liberty. The total attendance in the cave for the three summer months reached 647,830 visitors. The third most visited cave is the Demänovská Ice Cave. The total attendance in the cave for the three summer months 2010–2019 reached 600,510 visitors. The fourth most visited cave is the Dobšinská Ice Cave. The total attendance in the cave for the three summer months reached 578,766 visitors. Most visitors came to the Belianska cave in 2019, Demänovská Cave of Liberty in 2016, Demänovská and Dobšinská Ice Cave in 2017. The least visitors came to the caves in 2010, except of Demänovská Ice Cave (2011) when the cave was closed for most of July due to reconstruction works. The highest value was reached by visitors for the period 2010–2019 in all four caves in August, the lowest attendance was traditionally in June.

The second category (moderately visited caves) includes caves with total attendance during the three summer months for 2010–2019 in the range 140,000–500,000 visitors. The fifth most-visited cave is the only cave accessible in western Slovakia, the Driny Cave (249,930 visitors total). The highest attendance values were reached in August, the lowest attendance was unconventionally observed in July. Most visitors came to the cave in 2017, the least in 2010. The sixth most visited cave is the Domica Cave (210,262). The highest

value was reached by visitors untraditionally in June, the lowest attendance was in July. Most visitors came to the cave in 2016, the least in 2014. From the aspect of attendance, the Ochtinská Aragonite Cave is the seventh in a row (192,186). Most visitors came to the cave in 2019, the least in 2013. The eighth most visited cave is the Harmanecká Cave (147,624). Most visitors came to the cave in 2016, the least in 2010. The ninth most visited cave is the Bystrianska Cave (146,569). Most visitors came to the cave in 2018, the least in 2013. The highest number of visitors in a case of Ochtinská Aragonite Cave, Harmanecká Cave and Bystrianska Cave was reached in August, the lowest attendance was traditionally in June.

The third category (the least visited caves) includes caves with a total attendance during the three summer months for the period 2010–2019 in the interval below 140,000 visitors. The tenth most visited cave is the Jasovská Cave (117,059). Most visitors came to the cave in 2019, the least in 2010. The Važecká Cave (112,921) is the eleventh most visited cave in the Slovak Republic. Most visitors came to the cave in 2018, the least in 2012. The Gombasecká Cave holds the position of the least visited cave amongst all accessible caves in Slovakia (72,925 visitors total). Most visitors came to the cave in 2019, the least in 2013. The highest value was reached by visitors for the period 2010–2019 in all three caves in August, the lowest attendance was traditionally in June.

The total number of visitors during the three summer months for all 12 caves together (Table 4, Figure 3) shows that since 2010 the number of visitors has been steadily increasing, peaking in 2017 (426,251 visitors). In 2018, there was a slight decrease (419,487), and a slight increase was observed in 2019 (426,199). A more significant growth in the number of visitors to the caves is to be seen in 2016.

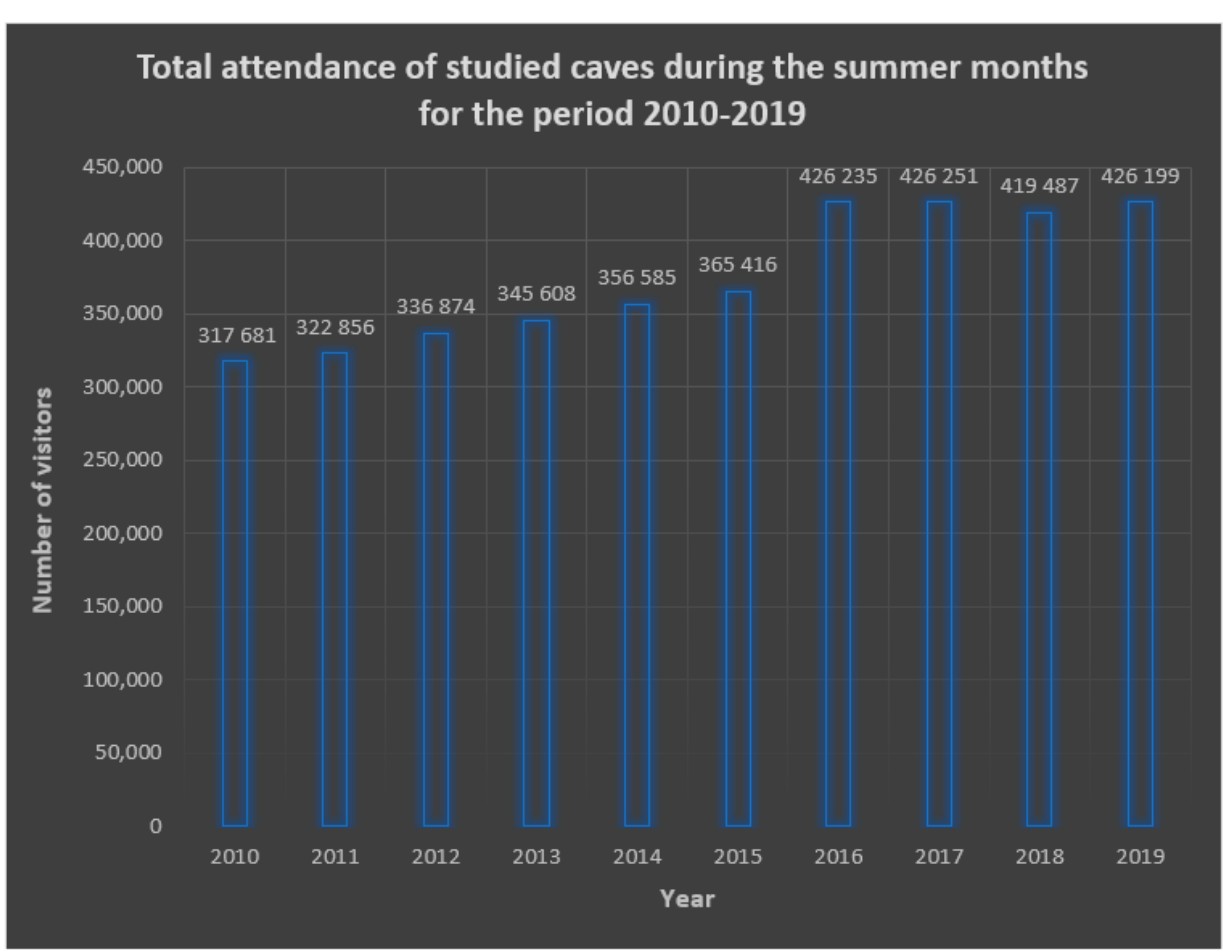

**Figure 3.** The total number of visitors to the studied caves for the summer months in 2010–2019.

Based on the overall result (Table 5, Figure 4) of the evaluation of the potential of caves for speleotourism (public opinion survey), we divided the studied caves into three categories (Figure 5):

**Table 5.** Potential values for individual classification traits and the total value of potential for speleotourism (K) of individual caves.

|  | 1 | 2 | 3 | 4 | 5 | 6 | 7 | 8 | 9 | 10 | 11 | 12 |
|---|---|---|---|---|---|---|---|---|---|---|---|---|
| $k_1$ | 41.392 | 55.536 | 34.424 | 28.912 | 62.192 | 59.952 | 49.504 | 53.04 | 55.224 | 55.774 | 52.832 | 51.168 |
| $k_2$ | 41.6 | 48.092 | 39.104 | 40.856 | 63.056 | 44.096 | 39.088 | 48.192 | 45.36 | 29.224 | 46.488 | 44.512 |
| $k_3$ | 23.608 | 54.184 | 29.432 | 56.472 | 56.68 | 56.368 | 49.816 | 52.728 | 25.168 | 50.336 | 36.816 | 43.888 |
| $k_4$ | 42.364 | 27.234 | 48.597 | 28.569 | 24.202 | 27.926 | 48.683 | 43.254 | 46.636 | 36.579 | 48.772 | 40.139 |
| $k_5$ | 32.841 | 35.6 | 18.067 | 21.538 | 51.4 | 45.6 | 34.71 | 18.423 | 17.8 | 53.4 | 48.861 | 34.532 |
| $k_6$ | 46.013 | 32.93 | 22.695 | 48.06 | 49.662 | 42.129 | 30.438 | 49.128 | 24.119 | 37.558 | 38.537 | 39.882 |
| $k_7$ | 23.616 | 14.592 | 38.272 | 18.504 | 21.952 | 24.192 | 38.4 | 21.632 | 26.496 | 38.4 | 38.08 | 37.12 |
| $k_8$ | 23.68 | 22.144 | 18.88 | 16.704 | 20.992 | 23.552 | 21.824 | 20.736 | 20.416 | 20.928 | 23.68 | 22.976 |
| $k_9$ | 34.368 | 29.44 | 32.152 | 33.91 | 34.624 | 33.984 | 34.048 | 25.312 | 35.2 | 31.744 | 35.328 | 32.386 |
| $k_{10}$ | 33.792 | 31.616 | 34.608 | 27.264 | 33.664 | 28.928 | 35.136 | 29.504 | 34.752 | 32.256 | 35.52 | 30.976 |
| $k_{11}$ | 13.248 | 13.504 | 15.808 | 35.776 | 13.312 | 12.8 | 36.544 | 13.888 | 12.8 | 36.928 | 13.824 | 30.464 |
| $k_{12}$ | 19.512 | 11.484 | 14.328 | 19.8 | 19.44 | 15.012 | 19.08 | 10.476 | 9.792 | 13.752 | 15.792 | 16.036 |
| $k_{13}$ | 14.112 | 11.376 | 8.712 | 15.127 | 12.452 | 8.676 | 10.247 | 9.795 | 7.043 | 16.071 | 12.475 | 13.276 |
| $k_{14}$ | 7.672 | 7.48 | 4.018 | 8.271 | 7.798 | 7.63 | 6.027 | 7.617 | 7.128 | 8.417 | 6.417 | 7.512 |
| $k_{15}$ | 6.272 | 4.144 | 6.636 | 3.721 | 7.602 | 7.224 | 4.758 | 6.028 | 6.545 | 7.028 | 5.171 | 6.831 |
| **K** | 404.09 | 399.356 | 365.733 | 403.484 | 479.028 | 438.069 | 458.303 | 409.753 | 374.479 | 468.395 | 458.593 | 451.698 |

Legend to Table 5: 1—Driny Cave, 2—Harmanecká Cave, 3—Bystrianska Cave, 4—Demänovská Ice Cave, 5—Demänovská Cave of Liberty, 6—Belianska Cave, 7—Važecká Cave, 8—Dobšinská Ice Cave, 9—Ochtinská Aragonite Cave, 10—Domica Cave, 11—Gombasecká Cave, 12—Jasovská Cave.

The first category (caves with the highest potential for speleotourism), (potential value over 435 points)—in terms of evaluating the public opinion survey, it can be stated that the Demänovská Cave of Liberty has the highest value of potential for speleotourism. Respondents positively assessed especially the rich occurrence of dripstone decoration along the entire tour route, the dimensions and spaciousness of cave corridors and domes, entertaining and detailed explanation of the guide, the occurrence of other tourist destinations within a distance of 10 km from the cave, and a good level of service at the cave entrance. On the contrary, respondents pointed out the difficulty of the tour route in terms of elevation and number of stairs and negatively assessed the difficulty of the access route from the parking lot. According to the respondents, the Domica Cave, which is part of the UNESCO World Natural Heritage List, has the second-highest potential for speleotourism. Respondents positively assessed the occurrence of lakes and underground watercourse with the possibility of navigation, seamless accessibility from the parking lot to the cave entrance, a rich occurrence of archeological finds and speleofauna, and an extensive educational exposition in the cave lobby. On the contrary, respondents pointed to the unprofessional, hasty approach of the guide and insufficient information during the interpretation. Third in the ranking of the evaluation of caves' potential for speleotourism is the Gombasecká Cave, which is also part of the UNESCO World Natural Heritage List. Respondents especially positively assessed an undemanding tour route without large elevations, entrance fee appropriate to the experience, lighting, and sufficient safety on the route. The cave received a relatively high score in terms of variety and attractiveness of the decoration, interpretation of the guide, ease of the tour route, the occurrence of watercourses and lakes, and accessibility from the parking lot to the cave entrance. According to the

respondents, the Važecká Cave has the fourth-highest potential for speleotourism. Respondents positively assessed the easy access from the parking lot to the cave entrance, entrance fee commensurate with the experience gained, good transport accessibility concerning the main transport routes, rich occurrence of paleontological finds (cave bear bones), lighting, and safety on the tour route. Another cave of the UNESCO World Natural Heritage List, the Jasovská Cave, has the fifth-highest potential for speleotourism. It maintains high point values for several indices, e.g., variety and attractiveness of the cave decoration, occurrence of underground lakes, accessibility from the parking lot to the cave entrance, entrance fee appropriate to the experience, lighting, and safety on the tour route, occurrence of archaeological finds and speleofauna. The last cave in this category, with the sixth-highest potential for speleotourism, is the Belianska Cave. It acquired high point values in terms of variety and attractiveness of the cave decoration, dimensions, and spaciousness of cave corridors and domes, the occurrence of underground lakes, entrance fee adequate to the experience and lighting of cave spaces.

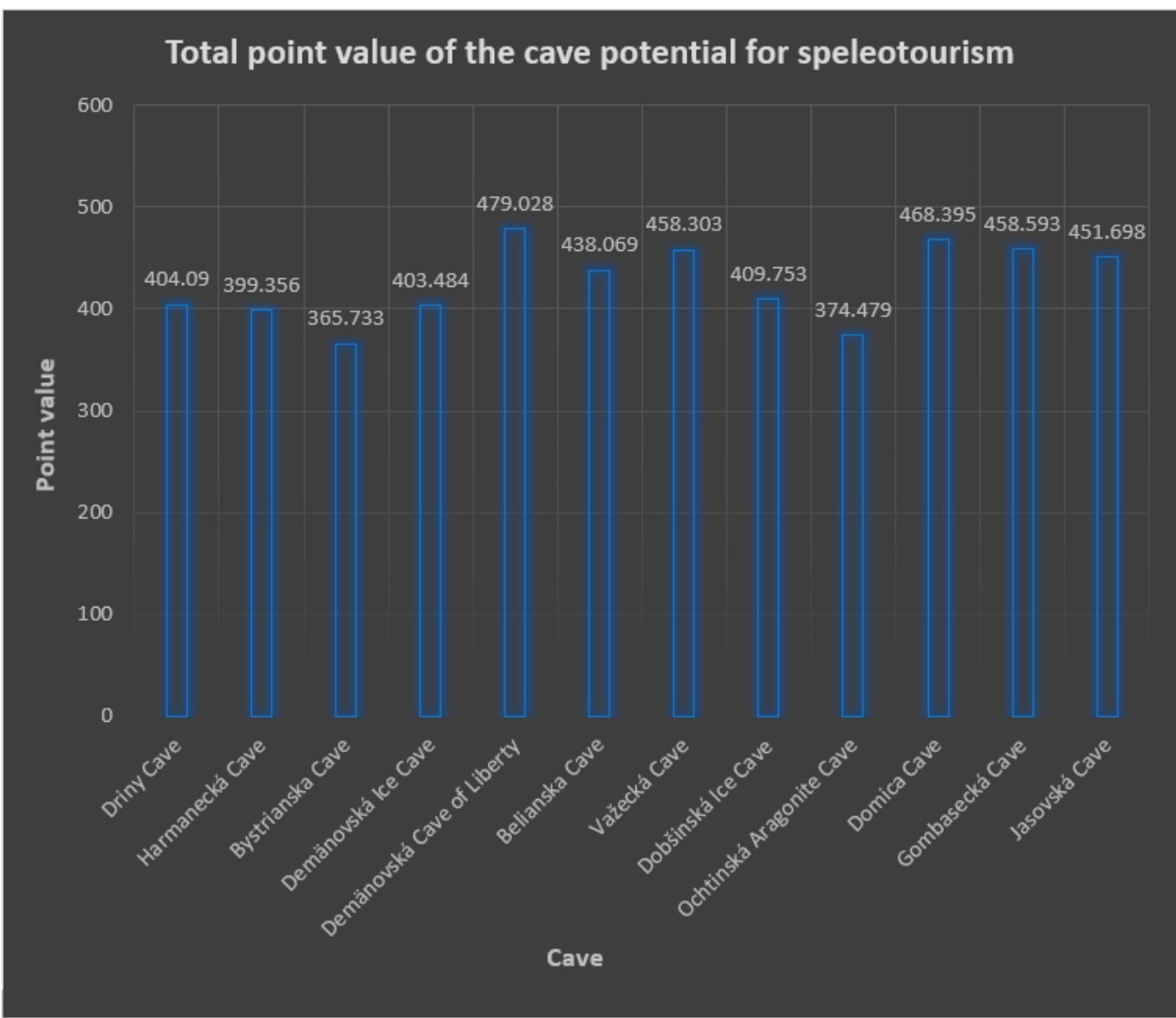

**Figure 4.** Total point value of cave potential for speleotourism.

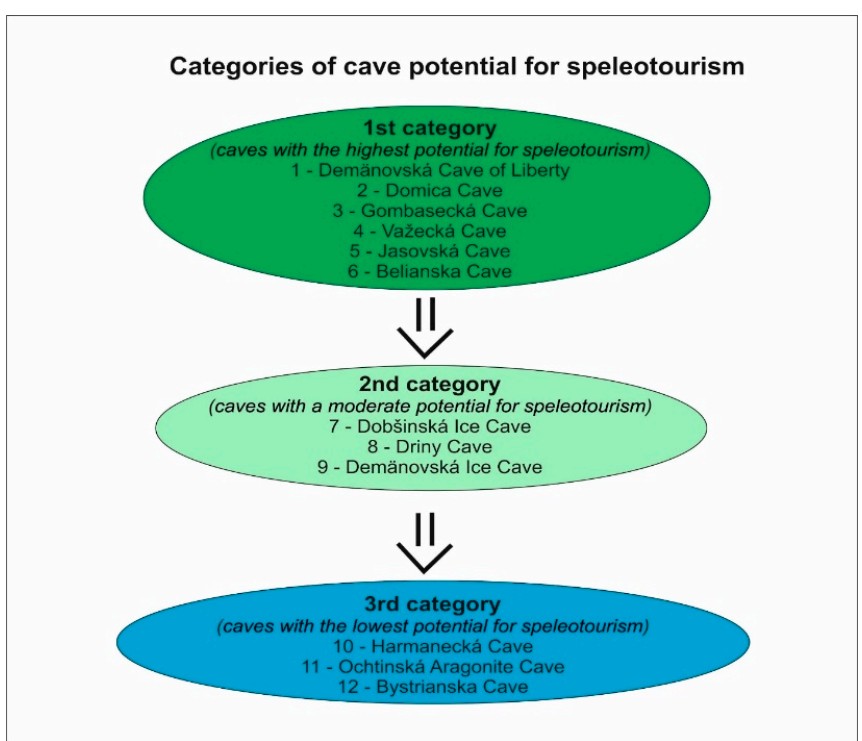

**Figure 5.** Categories of cave potential for speleotourism.

Within the second category (caves with a moderate potential for speleotourism —potential point value between 400 and 435 points)—the seventh-highest potential for speleotourism was acquired by the Dobšinská Ice Cave, which is a UNESCO World Natural Heritage Site. Respondents relatively highly rated the richness and volume of ice decoration, the interpretation of the guide, the dimension and spaciousness of the cave spaces, and the occurrence of other tourist destinations in the cave's vicinity. The lowest point rating was given to the cave at classification index 9—the respondents pointed to the cave spaces' low lighting. The eighth cave in a row is the Driny Cave. The cave received the highest point rating of all caves at classification index 8—the entrance fee commensurate with the gained experience. On the contrary, respondents pointed to relatively narrow, low areas of the cave. The last cave in this category (ninth in the ranking) is the Demänovská Ice Cave. Respondents positively assessed the excellent transport accessibility concerning the main transport routes. On the contrary, respondents pointed out the small variety and attractiveness of decoration (visit took place in June 2019, when ice decoration was minimal due to mild winter and other factors). The matter resulted in a negative evaluation of the entrance fee, which did not correspond to the experience gained. Safety on the tour route and poor service at the cave entrance was also perceived negatively. Due to the relatively high score for other indicators, the cave was eventually included in this second category.

The third category (caves with the lowest potential for speleotourism, the point value of potential is below 400 points)—the tenth cave in the ranking is the Harmanecká Cave. The cave received the lowest point rating of all caves at classification index 7 —respondents pointed to a long and demanding ascent from the parking lot to the cave entrance. Respondents rated the snow-white cave decoration, the size and size of the cave spaces, and the interpretation relatively high in points. The penultimate eleventh cave in the ranking is the Ochtinská aragonite cave, part of the UNESCO World Natural Heritage List. The cave received the lowest point rating of all caves at four classification indices. Respondents pointed to the absence of watercourse and lakes, the absence of archaeological and paleontological finds, the absence of speleofauna, and poor transport accessibility concerning major transport routes. The cave received a high score in terms of occurrence of a unique aragonite decoration. The simplicity of the tour route, lighting, and safety

was evaluated positively. The Bystrianska Cave became the cave with the lowest potential for speleotourism. There was the lowest-rated occurrence of other tourist destinations in the vicinity of the cave and information panels about the cave and the surrounding area. Respondents also pointed to the guide's insufficient interpretation, the absence of water flow and lakes on the tour route, insufficient lighting of some cave spaces, and lower transport accessibility concerning the main traffic routes. Respondents highly rated the tour route's unpretentiousness and access from the parking lot to the cave entrance.

## 4. Discussion

In terms of verification of the established research questions (hypotheses), we state the following facts:

1. We assumed the highest attendance of caves in 2019 is due to the continuously increasing number of tourists in Slovakia since 2014, culminating in 2019. This assumption has not been confirmed. Based on data from the Statistical Office of the Slovak Republic [88], it follows that the number of participants in a personal travel per year has been steadily increasing since 2014, peaking in 2019, when the number of residents aged 15 and over who participated in tourism for personal purposes reached the number of 3,413,879. In total, 6,432,934 people took part in tourism in Slovakia in 2019, a 15% increase compared to 2018. The number of foreign tourists reached the value of 2,475,094 in 2019, which is a 9.7% increase compared to 2018. These indicators were not reflected in the total attendance of the studied caves. The peak of the number of visitors to the studied caves (Table 4) in the three summer months occurred in 2017 (426,251 visitors). The second highest value was 2016 (426,235), the third 2019 (426,199). However, the difference between the year 2016 and 2017 is almost negligible, only 16 visitors. If we take into account the year-round attendance of caves, according to the Slovak Caves Administration, there is also the highest value of attendance in 2017 (633,158 visitors), followed by 2019 (628,433) and 2018 (621,852). In 2020, due to the coronavirus pandemic, the total number of visitors to the 12 studied caves reached 384,834. However, the caves were open in that year only from June 16 to October 2. Overall, we can say that the number of visitors to the caves does not copy the number of tourists.

2. We expected the largest number of visitors in the Demänovská Cave of Liberty and the Belianska Cave, due to the high concentration of tourism participants in the Demänovská Valley in the Low Tatras National Park in winter and summer, as well as in the Tatra National Park in the case of the Belianska Cave. This assumption has been confirmed. The most visited cave in Slovakia is Belianska Cave. The total attendance in the cave for the three summer months in 2010–2019 reached 703,240 visitors. According to the Slovak Caves Administration, the year-round attendance for 2019 reached 144,376 and is also the highest of all caves. The second most visited is the Demänovská Cave of Liberty. The total attendance in the cave for the three summer months in 2010–2019 reached 647,830 visitors. According to the Slovak Caves Administration, the year-round attendance for 2019 reached 118,703 visitors and is also the second-highest of all caves. Both caves are located in national parks, which are among the most visited in Slovakia. Demänovská Dolina, in which the Demänovská Cave of Liberty lies, is the most important tourist center in the Low Tatras National Park, experiencing high attendance in summer and winter, as it is also the largest ski resort in Slovakia, Demänovská Dolina-Jasná. The cave is open all year round, except for the period from November 16 to January 1. This significant location factor is also confirmed by the second accessible cave of the Demänovská valley—the Demänovská Ice Cave, which is the third most visited cave during the summer months of the period 2010–2019. Belianska Cave is one of two accessible caves in the Tatra National Park (the other is the Brestovská Cave, which opened to the public in 2016). Attractive tourist destinations in summer and winter (several ski

resorts) are situated nearby. The cave is increasingly visited even in bad weather and is also open all year round, except from November 16 to January 1.

3.  We assumed that the guide's approach and interpretation are essential factors in the cave's overall evaluation regarding its attractiveness, respectively, the potential for speleotourism. This assumption was confirmed. It turns out that the guide's approach, combined with the form of presentation and the content of the interpretation, is an essential factor in the overall perception of the cave's attractiveness and, thus, its overall potential for speleotourism evaluation. Respondents especially appreciated the precise, detailed explanation of the cave being presented in a light, humorous form with the opportunity to ask additional questions. The Demänovská Cave of Liberty received the highest point rating within this classification index, but a relatively high point rating was also awarded to several other caves that were less attractive in terms of dimensions or variety of decoration. Deficiencies in the variety or attractiveness of decoration, or the dimension and spaciousness of the cave spaces, resulting from the cave's very nature, can thus be compensated by the subjective factor of the guide's approach and interpretation. This fact is more significantly reflected in the final evaluation of the potential, as the classification index interpretation of the guide belongs to the first class of three classification indices with the highest value of the index weight. The need for a variable and varied offer of classification indices (criteria, indicators) with different weights of significance is pointed out in the published works, e.g., [82–85] and others.

4.  We assumed a correlation between the highest number of visitors and the highest value of the cave potential for speleotourism; similarly, the correlation between the lowest number of visitors and the lowest potential of the cave for speleotourism. This assumption was not confirmed; respectively, it was confirmed only partially. Jasovská Cave, Važecká Cave, and Gombasecká Cave belong to the third category in terms of attendance (least visited caves), but in terms of evaluating the speleotourism potential, they belong to the first category (caves with the highest potential for speleotourism). Belianska Cave is the most visited cave (first category of attendance). Within the category of speleotourism potential, it is also a part of the first category (caves with the highest potential for speleotourism), but in the point evaluation, it ranks only in the sixth place in this category. Demänovská Cave of Liberty is the second most visited cave, and it ranks first in the potential for speleotourism category, so the assumption of correlation in its case was confirmed to a greater extent than in the case of the Belianská Cave. This relative correlation discrepancy between the rate of attendance and the potential of the cave for speleotourism also points to other factors that enter into evaluating the attractiveness and potential, respectively, which are decisive for tourists' decision to visit the cave.

5.  We assumed that the Demänovská Cave of Liberty would have the most significant potential for speleotourism. This assumption was proved. The cave is often mentioned in the media, in various promotional materials, and book guides for the domestic and foreign markets. It is located in one of the most visited valleys in Slovakia, where tourism is distributed evenly throughout the year. The character of the cave supports the point evaluation of the potential. The cave has a varied and lively decoration on the whole tour route, dominated by massive and large spaces, underground lakes, and a watercourse. In our opinion, only the Domica Cave in the Slovak Karst can compete with it, which was also confirmed by its second-highest value of potential. The Domica Cave has a similar character of spaces as the Demänovská Cave of Liberty with rich and varied decoration, huge spaces, underground lakes, and a watercourse. In addition to the Demänovská Cave of Liberty, the Domica Cave also has the option of boating on an underground watercourse, the occurrence of archaeological finds, including paintings, and the UNESCO World Natural Heritage brand is not negligible.

## 5. Conclusions

Analysis and evaluation of cave attendance over a more extended period provides valuable information on development trends in this area and allows proper management and adoption of adequate measures for better marketing while respecting these caves' limits concerning nature conservation. Similarly, the analysis of these objects' potential based on a public opinion survey among visitors provides valuable data on their perception based on evaluating specific classification indices. These data can also be used in the management of the cave and its promotion. Determining the weight of individual classification indices provides insight into the importance of individual criteria in assessing caves' attractiveness for speleotourism.

In terms of the results of the presented study, the following most important findings can be stated:

1. The correlation between attendance and the potential for speleotourism was not confirmed. The highest number of visitors does not mean the highest potential for speleotourism (for example, Belianska Cave).
2. A relatively similar amount of cave attendance in the years 2016–2019 is partly due to limitations, namely the ceiling of attendance at several caves.
3. Variety and attractiveness of cave decoration, guide interpretation, and the size of cave corridors and domes are among the most important criteria (they have the highest importance), on the basis of which the respondents evaluate the attractiveness and potential of caves for speleotourism.
4. In the case of 5 caves, the UNESCO World Natural Heritage brand does not automatically mean the highest attendance, with respect to the highest potential for speleotourism.
5. As expected, the Demänovská Cave of Freedom received the highest value of potential for speleotourism and also has the second highest attendance. The cave is well promoted, it is located in one of the most visited localities and the character of the cave itself helps the overall evaluation.

This type of research has not previously been carried out in the geographical area of the Slovak Republic. It is an initial study that can be followed up on a larger scale in several areas in other studies. For example, an analysis of the trend in attendance at these caves in 2020—reflecting the coronavirus pandemic in this area—using a public opinion re-survey could be interesting.

The research results led to the following recommendations in relation to the sustainable development and management of show caves:

1. Pay attention to the strict selection of guides in the caves and their accompanying words, as the individual approach of the guide is considered by the respondents to be an important factor in the attractiveness and potential of the cave for speleotourism.
2. To regulate and monitor the impact of visitors in both ice caves, with regard to the decrease of ice decoration.
3. To ensure greater promotion of show caves in Slovakia at home and abroad, to bring them more into the public consciousness.
4. Implement the gradual renewal of the entrance areas of caves and sightseeing routes.
5. Review technical interventions in some caves, given the fragility and vulnerability of the cave geosystem, in order to make sustainable use of these areas.

The accessible caves in the Slovak Republic represent significant phenomena, with a specific and fragile natural and technical geosystem subject to strict nature protection, in the fifth highest degree of protection, with five of them belong to the UNESCO World Natural Heritage List. At the same time, they are the essential objects of speleotourism.

The criteria for evaluating the potential of caves for speleotourism have a general character and can be used for similar evaluations in other countries considering the presence of show caves with built tourist infrastructure.

Moreover, harmonizing nature protection requirements with the aim of making their underground beauty available to the civil and professional public is the main task of the Administration of Slovak Caves in the future.

**Author Contributions:** Conceptualization, V.Č. and B.G.; methodology, V.Č.; software, V.K.; validation, V.Č., R.K., P.H. and B.G.; formal analysis, P.C.; investigation, V.Č.; resources, R.K.; data curation, V.K. and P.H.; writing—original draft preparation, V.Č.; writing—review and editing, V.Č.; visualization, V.K.; supervision, B.G.; project administration, B.G.; funding acquisition, B.G. All authors have read and agreed to the published version of the manuscript.

**Funding:** This research and the APC was funded by SLOVAK RESEARCH AND DEVELOPMENT AGENCY, grant number APVV-18-0185 "Transformácia využívania kultúrnej krajiny Slovenska a predikcia jej ďalšieho vývoja" and also by RESEARCH GRANT AGENCY, grant number 1/0667/21 "Environmentálne špecifiká životného prostredia vybraných montánnych vodohospodárskych systémov na Slovensku".

**Institutional Review Board Statement:** The data collection procedure was accomplished in accordance with the guidelines of the Declaration of Helsinki for the protection of human research subjects.

**Informed Consent Statement:** Informed consent was obtained from all subjects involved in the study.

**Data Availability Statement:** The data presented in this study are available on request from the corresponding author.

**Acknowledgments:** The authors express their sincere thanks to Miroslav Kudla from the Slovak Caves Administration in Liptovský Mikuláš for providing data on the number of visitors to the show caves of Slovakia.

**Conflicts of Interest:** The authors declare no conflict of interest.

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
