# Peer review of "Analysis of Attendance and Speleotourism Potential of Accessible Caves in Karst Landscape of Slovakia"

_sustainability, doi:10.3390/su13115881_

Round 1
Reviewer 1 Report
Thank you for the opportunity to review your manuscript. The issue of sustainable speleotourism development is an important one, and I appreciate the authors' interest in this subject.
I have attached an annotated PDF with numerous comments evidencing specific suggestions, comments, and concerns with the article as it is currently presented. While I will not share all of those here, the most important are the following:
- This article is largely atheoretical and does justify the measures for "potential" within the literature. I encourage the authors to review the recreation suitability mapping and indicators of quality literature for examples how to connect specific items with constructs that inform the outcome variable of "potential."
- Related, the weighting scheme is not clear. What are the theoretical drivers that informed placement into different classifications?
- There are no measures of statistical significance, effect sizes, etc. present. As a result descriptive trends are referred to in order to evidence points, but its unclear of those trends actually matter statistically or practically.
- There is no information about human subjects protocol and protection of the research participants. Since these folks were students, there are a variety of ethical concerns with their blinded participation and potential coercion.
- Further, deriving a ranking based on student perceptions has obvious methodological issues none of which are mentioned by the authors. Not only could these students be influenced by their instructors, who appear to be the authors of this paper, but they are a relatively homogeneous population, so the generalizability of the results - irrespective of them being potentially biased by their coursework of instructor perceptions - is tenuous at best.
- Structurally, there are many pages, figures, and tables that are well written, but are unnecessary. The authors' job is reduce data to inform our knowledge in a parsimonious way. I have noted multiple locations where this is necessary in the PDF.
- Finally, while certain caves are mentioned as UNESCO World Heritage Sites, the authors do not make any connection with sustainability themes and concepts explicit. As a result, this article seems more suited for tourism or recreation journals.
It is possible that certain information missing or unclear in the article could be revised and added to addresses the comments I mentioned above. If so, I encourage the authors to submit a heavily revised version of this article to a tourism or recreation journal, such as Journal of Sustainable Tourism, Journal of Outdoor Recreation and Tourism, etc., as their subject is important. That said, as currently written, I do not believe it is suitable for publication in Sustainability.

Author Response
Please see the attachment.
Comments on notes are written as the answer in the manuscript.

Reviewer 2 Report
Dear Authors, I am missing the sustainability approach in your research as caves are sensitive systems and can be impacted negatively by the increasing number of visitors (especially in the ice caves). In your research, you are focusing in detail on caves; however, the analysis of the speleotourism potential in Slovakia is insufficient. It is not justified why you choose the criteria on page 6. The study lacks examples of studies in this field to compare the reliability of the criteria. I recommend shortening the study.
Reviewer 3 Report
This paper very interestingly uses the criteria for describing caves used as case study giving an excelent insight in their potential for speleotourism (cave attendance category). The only objection is that the sample was made up of people with education related to cave valuation, and that the results would certainly be significantly different for the general population.
The overall research is very well structured: literature review is comprehensive and up-to-date; methodologically it is sound and robust; discussion is insightful and conclusions are very well elaborated.
Round 2
Reviewer 1 Report
Thank you for the opportunity to review your revised manuscript. The issue of sustainable speleotourism development continues to be an important one, and I appreciate the authors' attention to a portion of suggested revisions.
I have attached an second annotated PDF with numerous comments evidencing specific suggestions, comments, and concerns with the article as it is currently presented. While I will not share all of those here, the most important are the following:
- This article continues to be largely atheoretical and does justify the measures for "potential" within the literature. There has been a good revision of the front section which includes information from the caving literature. That said, there needs to be some kind of psychological, sociological, etc. theory base to ground the key criteria of quality in. It's mentioned that these are created and gleaned from previous research in other settings, but again we need theoretical grounding of each of them to understand why they are important to the visitor experience.
- There continue to be no measures of statistical significance, effect sizes, etc. present. As a result descriptive trends are referred to in order to evidence points, but its unclear of those trends actually matter statistically or practically.
- There has been some good clarification about who is in the sample, however, that limitation needs to be returned to heavily at the end in a limitations section. And, frankly, I'm not buying the voluntary nature of the student surveys. Were they truly voluntary if this was part of their field practice? More information is needed here to convince readers that this was truly voluntary and students, who comprise the majority of your respondents, could opt out.
- Structurally, there continue to be many pages, figures, and tables that are well written, but are unnecessary. The authors' job is reduce data to inform our knowledge in a parsimonious way. I have noted multiple locations where things could be moved into appendixes, represented together or removed. Perhaps more importantly, statistical analyses will help replace some of these raw data sums to more meaningful useful data
- Finally, the sustainable development argument of the cave system is definitely improved in this revision. Nice job! That said, I think you can leverage the UNESCO World Heritage Site designation further in the introduction and connect those to sustainability more explicitly with a simple addition of a few sentences.
With statistical analyses, improved data representation, and theoretical grounding, this paper may be suitable for publication. That said, as currently written, I still do not believe it is suitable for publication in Sustainability.

Author Response
reply was sent.

Reviewer 2 Report
Dear Authors, I am still missing the sustainability approach in your research. You did not look at the visitation of the caves according to the Limits to Acceptable Change Model, the specially developed model for protected areas, determining the acceptable burden on the territory through the use of indicators of sustainable tourism development. It is a general model of planning and regulation of influences, especially on the biosphere and social environment. You do not deal with the absorption capacity of the destination, e. g. Physically carrying capacity, Ecologically carrying capacity, and Socio-cultural carrying capacity. The study lacks examples of studies in this field to compare the reliability of the criteria - why did you choose exactly these presented 15 criteria.
Author Response
Reply was sent.
